# LoQT: Low Rank Adapters for Quantized Training

Sebastian Loeschcke [* 1 2]   Mads Toftrup [* 3]   Michael J. Kastoryano [1]   Serge Belongie [1]   Vésteinn Snæbjarnarson [1]

## Abstract

Training of large neural networks requires significant computational resources. Despite advances using low-rank adapters and quantization, pretraining of models such as LLMs on consumer hardware has not been possible without model sharding, offloading during training, or per-layer gradient updates. To address these limitations, we propose LoQT, a method for efficiently training quantized models. LoQT uses gradient-based tensor factorization to initialize low-rank trainable weight matrices that are periodically merged into quantized full-rank weight matrices. Our approach is suitable for both pretraining and fine-tuning models, achieving similar performance to full training, which we demonstrate experimentally for language modeling and downstream task adaptation. We find that LoQT enables efficient training of models up to 13B parameters on a consumer-grade 24GB GPU.

## 1. Introduction

Training large neural networks requires substantial hardware and energy resources. Reducing these requirements is thus important for cost efficiency and environmental sustainability, while also lowering the entry barrier for researchers and practitioners. The main barriers in training large models are the compute operations required, as well as the memory needed to store those computations, in this paper we focus on the latter. Memory use during training comes primarily from the weights of the model itself as well as the optimizer states used to train the model. To target the weights, variations on low-rank adaptation (LoRA) (Hu et al., 2021; Hayou et al., 2024b; Dettmers et al., 2023a; Lialin et al., 2023; Liao & Monz, 2024) have been suggested to decrease the number of trainable parameters, in combination with the use of low precision representations. To target the optimizer,

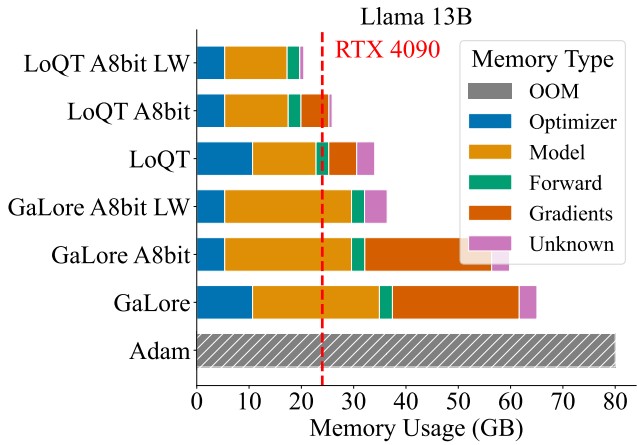

Figure 1: Memory usage of Llama 13B, rank 1024. LW: per-layer gradient updates. A8bit: Adam 8-bit

low-rank approaches for projecting gradients to a lower rank have been employed (Zhao et al., 2024). Finally, various applications of quantization (Gholami et al., 2021; Ma et al., 2024; Dettmers et al., 2023a) have been used to decrease memory requirements. In this work, we combine and extend these approaches into a highly memory-efficient training configuration.

In typical training setups, the optimizer states take up larger space than the model itself, as methods such as Adam (Kingma & Ba, 2017) need to keep track of two parameters for each weight of the model. GaLore (Zhao et al., 2024) significantly reduces the number of parameters needed for storing the optimizer states by only keeping track of the optimizer state in a low-rank projection, which is then projected up to be applied to the model weights. Combining this method with quantization would further shrink the footprint of the model. However, updating the weights of a highly quantized model directly in low-precision space has not been shown to work. This is mainly due to the higher-precision gradient updates being too small to have an impact on the lower-precision quantized states. Lastly, while LoRA is memory efficient for parameter-efficient fine-tuning of pre-trained models, it does not work as a pretraining method by itself (Lialin et al., 2023).

To address these shortcomings, we propose a new method, LoQT. LoQT initializes two low-rank factors for each

---

[*]Equal contribution [1]University of Copenhagen [2]IT University of Copenhagen [3]Aarhus University. Correspondence to: Sebastian Loeschcke <sbl@di.ku.dk>, Mads Toftrup <toftrup@cs.au.dk>.

Accepted to the Workshop on Advancing Neural Network Training at International Conference on Machine Learning (WANT@ICML 2024).

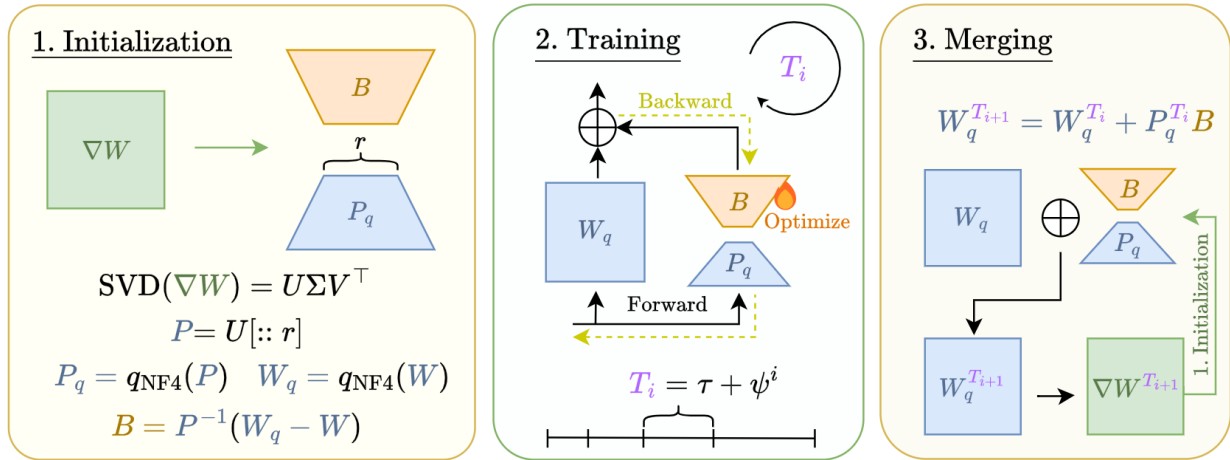

Figure 2: Overview of LoQT. (1) Low-rank factors $P$ and $B$ are periodically initialized from the gradient of the dequantized model weights $\nabla W$, (2) then only $B$ is trained while $P_q$ and $W_q$ are kept quantized and frozen, over an exponentially increasing interval until $T_i$, (3) the low-rank factors are merged back into the quantized model. The process is repeated until training halts.

weight matrix $W$: 1) $P$, initialized using a projection of $W$'s gradients into a low-rank subspace, and 2) $B$, which is initialized to minimize the quantization error. $B$ is the only directly trained matrix, which means that the optimizer state can be shrunk significantly. The product $PB$ is then periodically merged into the full rank matrix $W$ with exponentially increasing scheduling. As $W$ and $P$ do not receive gradient updates, they can be quantized, thus optimizing memory usage even further. The large accumulated updates make it possible to update a quantized model, as the addition of smaller changes would not register in the quantized state. A high-level overview is given in Fig. 2.

We show that LoQT works well with and without a quantized model, enabling not only a lower memory footprint in the optimizer state but also over the model parameters. Our results show that we get competitive performance to prior methods using significantly less memory, in particular when using quantization. We also demonstrate superior performance when fine-tuning pre-trained models by training and evaluating on the GLUE (Wang et al., 2018a) benchmark for natural language understanding. Finally, we ablate several properties of the suggested approach, and we find that an exponentially increasing projection gap is beneficial, not only to our work but also for prior work (Zhao et al., 2024). This is particularly crucial for the training of quantized models. LoQT enables efficient training of 7B models on consumer-grade hardware with 24GB of memory, and makes it feasible to train models with up to 13 billion parameters, without model parallel or by making use of per-layer gradient updates (Lv et al., 2023) on the same hardware as shown in Fig. 1.

## 2. Related Work and Background

### 2.1. Neural Network Quantization and NF4

Quantization compresses neural networks by converting high-precision values into lower-precision formats, significantly reducing storage requirements (Zafrir et al., 2019; Shen et al., 2019; Bai et al., 2021; Dettmers et al., 2022a). The process involves taking a datatype of high precision, such as 32-bit, requiring 4 bytes of memory, and converting it into a representation with increasing rounding errors but lower memory cost. In this work, we use NF4 quantization (Dettmers et al., 2023a). Since it is a 4-bit code, it can only represent $2^4$ different values. NF4 works by first normalizing values onto the interval $[-1 : 1]$. These are then discretized onto quantiles of the normal distribution, $(q_i)_{i=1}^{16}$ (see (Dettmers et al., 2023a) for details). The elements of a layer are divided into blocks of 64 weights. Each block $\beta$ has a scaling factor $\mathcal{M}_\beta = \max_{w \in \beta} |w_{32}|$.

$$w_{\text{NF4}} = q_{\text{NF4}}(w, \mathcal{M}_\beta) := \text{argmin}_{q_i} |w/\mathcal{M}_\beta - q_i|, \quad (1)$$

$$w = q_{\text{NF4}}^{-1}(w_{\text{NF4}}, \mathcal{M}_\beta) := \mathcal{M}_\beta \cdot w_{\text{NF4}}. \quad (2)$$

We provide an overview of different categories of quantization techniques, and how they relate to LoQT, in Appendix A. Compared to prior approaches, LoQT retains the benefits of reduced memory usage while minimizing accuracy loss, using high-precision updates on a low-rank representation. This allows for efficient model updates without the overhead of full matrix storage and re-quantization.

### 2.2. Adaptation of Pretrained Networks

Low-Rank Adaptation (LoRA) (Hu et al., 2021) enables fine-tuning of pre-trained models using low-rank adaptors,

effectively reducing the memory footprint by only training weight adaptors for targeted layers. However, simple low-rank training using LoRA factor matrices has not been shown to work for pre-training (Lialin et al., 2023).

LoRA employs trainable low-rank matrices $A$ and $B$ that are used to update $W$ following $W_t = W_{t-1} + AB$, where $W_{t-1}$ is frozen to enable precise adjustments within a low-rank framework. Extending this, LoRA+ (Hayou et al., 2024a) applies different learning rates to $B$ compared to $A$ to optimize tuning. LoRA-FA (Zhang et al., 2023) further simplifies the process by training only $B$, eliminating the need to store full-rank activations for efficient fine-tuning. DoRA (Liu et al., 2024) divides the LoRA adapter into independent magnitude and direction components, providing granular control over weight adjustments. PISSA initializes $A$ and $B$ using SVD of $W$ for faster convergence (Meng et al., 2024) and does not use gradient information $\nabla W$. Since LoRA only trains $A$ and $B$ and keeps $W$ fixed, QLoRA (Hu et al., 2021) explore quantizing $W$. They fine-tune a quantized model $q(W) = W_q$ with 4-bit precision using randomly initialized 16-bit precision factors $A$ and $B$. To address quantization errors $\mathcal{E} = |W_q - W|$, low-rank factors of the quantization error $\mathcal{E}$ have been used (Li et al., 2023).

LoQT extends LoRA to both pretraining and fine-tuning. Unlike traditional LoRA, LoQT uses $A$ and $B$ to refine $W$ throughout training, with $A$ initialized from $W$'s gradient projection and $B$ trained along this gradient path. LoQT also incorporates quantization and targeted optimization iterations similar in spirit to LoftQ (Li et al., 2023), correcting for quantization errors in $W_q$, thus better aligning it with the original non-quantized $W$.

### 2.3. Memory Efficient Optimization

**Optimizer memory consumption**    A significant portion of the memory needed to train neural networks is typically consumed by optimizer states. Notably, Adam (Kingma & Ba, 2017), one of the most widely used optimizers, uses double the amount of memory as the gradient matrix to maintain first and second-order gradient statistics. Efforts to reduce this overhead have led to the development of adaptive optimization algorithms like Adafactor (Shazeer & Stern, 2018), which achieves sub-linear memory costs by factorizing the second-order statistics into a row-column outer product. Ga-Lore (Zhao et al., 2024) expands on this concept by using low-rank factorization and projecting low-rank gradients up to a full-rank when updating model weights.

**Periodic updating of weight matrices**    ReLoRA (Lialin et al., 2023) combines low-rank updates with initial full-rank training. They find that doing one-third of the training in full-rank, and the subsequent two-thirds in low-rank (see

§2.2) results in comparable performance to standard training methods.

**Low-rank gradients**    GaLore (Zhao et al., 2024), focuses on the structure of the gradients, projecting them into a low-rank space using factors $P$ and $Q$, which are derived from a truncated singular value decomposition (SVD) of the weight matrix gradient, $G_W \approx P_r \Sigma_r Q_r$. This reduces memory costs associated with storing the optimizer states and aligns with findings from recent studies which suggest that learning primarily occurs within a low-dimensional subspace at a given time (Larsen et al., 2022; Gur-Ari et al., 2018). This can be further combined with applying per-layer gradient updates, reducing the memory needed for storing the gradients for the full model at once (Lv et al., 2023).

LoQT builds on GaLore's gradient projection (§3.1) to initialize LoRA factors while updating the full matrix following a schedule inspired by ReLora, while only training one low-rank matrix per layer. We achieve comparable quality to GaLore and better performance than ReLoRA while reducing tunable parameters and memory usage compared to both approaches.

## 3. Efficient Pretraining With LoQT

LoQT works by initializing and training low-rank adapters obtained by taking the SVD of a given layer's gradients. Let $W$ indicate the full weights matrix of a given layer, $P$ be the left factor constructed from the SVD decomposition of the gradients matrix: $\nabla W = U \Sigma V^\top$; i.e. $P$ consists of the first $r$ columns of $U$ corresponding to the singular vectors with the $r$ largest singular values of $W$. The update rule for an interval $[T_{i-1}, T_i]$ is then given by $W_{T_i} = W_{T_{i-1}} + PB$, where only the weights of $B$ are updated. $P$ and $W_{T_{i-1}}$ are kept constant over the time interval. We describe this in more detail below, followed by a discussion on periodic updating of the factor $P$, the enabling of quantized pre-training, error compensation, and exponential update intervals. Pseudo-code for LoQT is shown in Fig. 3.

### 3.1. Background: GaLore

Zhao et al. (Zhao et al., 2024) show that gradients exhibit a low-rank structure during training. They exploit this insight by projecting the gradient to a low-rank subspace and applying the Adam optimizer before projecting back to the original dimensions. By doing this, the memory-intensive optimizer states required by Adam are shrunk significantly for low enough ranks.

**Definition 3.1** (Gradient Low-rank Projection, def. 3.4 in (Zhao et al., 2024)).    Gradient low-rank projection (**GaLore**) denotes the following gradient update rules, where $\eta$ is the learning rate, $\rho$ is the Adam optimizer, and $W \in R^{m \times n}$ is the weight matrix being trained, and $T$ represents the

total number of training iterations between recomputing the projection matrices:

$$W_T = W_0 + \eta \sum_{t=0}^{T-1} \tilde{G}_t, \text{ where } \tilde{G}_t = P_t \rho_t (P_t^\top G_t Q_t) Q_t^\top,$$

where $P_t \in R^{m \times r}$ and $Q_t \in R^{n \times r}$ are are the top-$r$ singular vectors from the SVD decomposition of the gradient matrix at each iteration $t$. In practice, this can be approximated by only applying a one-sided projection, as in

$$W_T' = W_0 + \eta \sum_{t=0}^{T-1} P_t \rho_t (P_t^\top G_t) \quad \text{or} \tag{3}$$

$$W_T' = W_0 + \eta \sum_{t=0}^{T-1} \rho_t (G_t Q_t) Q_t^\top. \tag{4}$$

GaLore demonstrates that it is sufficient to keep the projection matrix fixed and only update it once every $T$ iterations, which we use in the following.

### 3.2. Low-rank Gradients as Adapters

We now describe the process by which we initialize the parameters we optimize in LoQT. We adopt the memory-performance trade-off of using only a one-sided projection. We compute $P^\top G$ if $m \le n$ and $GQ$ otherwise. We want to achieve a separation between trainable weights and static weights, which we achieve by rewriting GaLore in terms of low-rank adaptors. Assume, without loss of generality, that $m \le n$. Using the fact that $P_t$ is fixed in the interval $[0, T]$ we have that

$$W_T = W_0 + \eta \sum_{t=0}^{T-1} P \rho_t (P^\top G_t) \tag{5}$$

$$= W_0 + \eta \underbrace{P}_{\in \mathbb{R}^{m \times r}} \underbrace{\sum_{t=0}^{T-1} \rho (P^\top G_t)}_{B \in \mathbb{R}^{r \times n}} \tag{6}$$

From (4) it is clear that we can keep track of low-rank updates using rank-$r$ low-rank adaptors. We note that in the interval $[0, T]$ only $B$ is updated, creating the desired separation. If implemented directly, we would need to compute the gradient with respect to $W$ and then project it down using $P^\top G_t$. We find that this step is unnecessary; it is sufficient to train $B$ using standard gradient descent.

We now show that training the $B$ matrix using gradient descent is equivalent to training w.r.t. $W_t$ as in definition 3.1. Let $G^W$ indicate the gradient of the loss with respect to $W$, and $G^B$ for the gradient of the loss with respect to $B$. Given a weight matrix $W$, a factor $P$ and a matrix $B$, when computing the forward pass $y = xW + xPB$, the gradient

of a loss function $\mathcal{L}$ w.r.t. $B$ is $G^B = P^\top G^W$. This can be seen by applying the chain rule to get $G^W = x^\top \frac{\partial \mathcal{L}}{\partial y}$. The vector multiplied onto $B$ is $xP$ giving $G^B = (xP)^\top \frac{\partial \mathcal{L}}{\partial y} = P^\top x^\top \frac{\partial L}{\partial y} = P^\top G^w$. This shows that calculating the gradient w.r.t $B$ gives the same as projecting the gradient w.r.t $W$. It is thus clear that GaLore's low-rank gradient updates should be the same as those obtained using backpropagation through LoRA.

### 3.3. Enabling pretraining with LoRA

Previous work has shown that training low-rank weight matrices works well for fine-tuning pre-trained weights. However, it has been shown that training low-rank factors, and periodically merging them into frozen $W$, does not work when starting with a randomly initialized matrix (Lialin et al., 2023). Here we address this shortcoming to enable full training using low-rank weight matrices.

Inspired by prior work (Lialin et al., 2023; Zhao et al., 2024), we periodically update a given layer $W_{T+1} = W_T + P_T B_T$ at fixed steps $T \in \mathcal{T}$. This approach allows $W$ to evolve as a sum of low-rank matrices aligning with GaLore's strategy of updating the gradient subspace during training:

$$W_t = W_0 + \Delta W_{T_1} + \Delta W_{T_2} + \ldots + \Delta W_{T_n},$$

where $t = \sum_{i=1}^{|\mathcal{T}|} T_i$ and $\Delta W_{T_i} = P_{T_i} B_{T_i}$ represents the product of the learning from $B$ during the interval $T_i - T_{i-1}$ scaled by the learning rate $\eta$ and modulated by the gradient projection matrix $P_{T_i}$. After each update at iteration $T_i \in \mathcal{T}$, we reinitialize the low-rank factors $P_T$ and $B_T$. As in (Zhao et al., 2024), we compute the gradient of $W_T$ over a single batch, focusing only on $\nabla W_T$ without needing full optimizer states. Not requiring optimizer states reduces the memory increase compared to full-rank training.

With the updated $W_t$ and reinitialized $P_t$ and $B_t$, a new gradient subspace is established for exploring the next $T_{i+1} - T_i$ steps. Our method treats $W_t$ as the full-rank repository of accumulated updates. Although it is periodically updated, $W_t$ is not part of the optimizer state computations, and the gradients during the single forward pass are offloaded to cpu. Since the SVD calculations are done layerwise only the current layer is needed on GPU, or the SVD can be calculated on CPU. $P_t$ defines the general gradient subspace and trajectory for the upcoming $T_{i+1} - T_i$ steps, and $B_t$ is adjusted to navigate optimally within the direction set by $P_t$. As only $B_t$ is trained, the number of parameters needing optimizer states is drastically reduced.

### 3.4. Quantized Training

Given that $B$ is the only matrix accumulating gradients and undergoing changes, the other matrices $W$ and $P$ can be kept quantized. This approach allows storing the weights

in NF4 precision without requiring high-precision gradient and weights to update $W$ and $P$. This approach reduces the memory required for these matrices by almost a factor of 4. To the best of our knowledge, we are the first to enable efficient 4-bit quantized training using gradient descent without storing the weights in full precision.

We quantize weights $q_{\text{NF4}}(W) = W_q$ and $q_{\text{NF4}}(P) = P_q$ as described in §2.1. During periodic updates at interval time steps $(\sum_{i=1}^{n} T_i)_{n=1}^{\max}$, $P_q$ and $W_q$ are dequantized using the inverse function, $P_{BF16} = q_{\text{NF4}}^{-1}(P_{\text{NF4}})$ and $W_{BF16} = q_{\text{NF4}}^{-1}(W_{\text{NF4}})$. After this, $W_{T_i} = W_{T_{i-1}} + P_{T_{i-1}} B_{T_{i-1}}$ is computed and quantized. The quantization and dequantization processes are applied layer by layer, ensuring that not all layers are simultaneously in a non-quantized state to reduce memory usage. Moreover, the quantization state itself is re-quantized for further efficiency following (Dettmers et al., 2023a). We implement LoQT using weight-only quantization, meaning the quantized weights are loaded into memory and then dequantized before computing the matrix multiplications.

### 3.5. Compensating for Errors Introduced by Quantization

As the quantization process inevitably leads to a discrepancy between the non-quantized and quantized versions of $W$ we wish to reduce this effect as much as possible. While compensating for quantization errors has been done before (Li et al., 2023), we need a tailored solution for LoQT.

During the merging phase, we first dequantize to obtain $W_{T-1}$ and $P_{T-1}$, and then compute the update $W_T = W_{T-1} + P_{T-1} B_{T-1}$. This is immediately followed by re-quantizing to get $Q_T = q_{\text{NF4}}(W_T)$. Our goal is thus to minimize the quantization error $\|(Q_T + P_T B_T) - W_T\|_F$. The projection matrix $P_t$ is calculated as described in section 3.1, whereas we are free to choose $B_t$ to minimize the quantization error. To achieve this, we solve for $B_T$, giving $B_T := P_T^+(Q_T - W_T)$, where $P_T^+$ is the Moore-Penrose pseudo-inverse. Inspired by (Li et al., 2023) we then iteratively refine $B_T$, by recomputing $Q_T = q_{\text{NF4}}(W_T - P_T B_T)$ and recomputing $B_t$, improving the alignment between the full-precision $W$ and its quantized state.

As training advances and the learning rate decays, the magnitude of the update $B_{T-1}$ to form $W_T$ decreases. This leads to negligible differences between $|q(Q_t + P_t B_t) - Q_t|$, which results in the weights plateauing early, as depicted in Fig. 4a. To address this, we implement an exponentially increasing scheduler for updating $W$. Drawing from GaLore's observation on the exponential decay of gradient rank (Lemma 3.1 (Zhao et al., 2024)), we start with a frequency gap $\tau$ and progressively increase the update intervals by a factor of $\psi$. The sequence of updates is then given by $(T_i)_{i=0}^{\infty} = (\tau + \psi^i)_{i=0}^{\infty}$ Each $T_i$ marks a training

---

**Algorithm 1** LoQT: Low Rank Adapters for Quantized Training

---

**Require:** $W$: Weight, $T$: Update steps, $\eta$: LR, $r$: rank, $q_N(\cdot)$: N-bit quantization function.
1: $G_W \leftarrow \nabla_W \mathcal{L}(W)$
2: $W_Q, P_Q, B \leftarrow \text{Initialize}(W, G_W)$
3: **for** each $t$ in training steps **do**
4:     **if** $t \in T$ **then**
5:         $W \leftarrow W_Q + s \cdot P_Q \cdot B_t$
6:         $G^W \leftarrow \nabla_W \mathcal{L}(W)$
7:         $W_Q, P_Q, B_t \leftarrow \text{Initialize}(W, G^W)$
8:     **else**
9:         $B_{t+1} \leftarrow B_t - \rho(G_t^B)$
10:    **end if**
11: **end for**
12: **return** $\theta$

---

**Algorithm 2** Initialization Procedure

---

1: **Initialize**$(W, G^W)$:
2: $U, S, V^T \leftarrow \text{SVD}(G^W)$
3: $P \leftarrow U[:, :r]$ {First $r$ singular vectors}
4: $P_q \leftarrow q_N(P)$
5: $B \leftarrow 0$
6: $\hat{W} \leftarrow W$
7: **for** each $c$ in compensation steps $C$ **do**
8:     $Q_c \leftarrow q_N(\hat{W})$
9:     $B \leftarrow P^+(\hat{W} - Q_c)$
10:    $\hat{W} \leftarrow W - PB$
11: **end for**
12: **return** $Q_c, B, P_q$

---

Figure 3: Pseudo-code for LoQT.

step $t$ when $W$ is updated. This scheduling ensures more frequent updates earlier in training and more well-spaced adjustments later, allowing more accumulated gradients before each update.

## 4. Experiments and Results

### 4.1. Experimental Setup

We evaluate LoQT by training LLaMA-based (Touvron et al., 2023) language models on the C4 dataset (Raffel et al., 2019), a collection of processed text in English that was scraped from the internet (Raffel et al., 2019). We train models of sizes of 60M, 130M, 350M, and 1B parameters, adhering to single-epoch training cycles determined by Chinchilla Scaling Laws (Hoffmann et al., 2022). While LoQT is capable of training models up to 13 billion parameters on consumer GPUs, compute limits prevent us from training to convergence for sizes above 1B. We also benchmark LoQT on the GLUE test-suite for natural language

Table 1: Comparison of low-rank pre-training methods for LLaMA2-style language models on the C4 dataset. The table shows validation perplexity and memory estimates for model weights and optimizer states. The rank ratio $r/d_{model}$ is relative to the largest weight matrix dimension. Perplexity values are averaged over three seeds showing mean and standard error. (*) Denotes results from GaLore (Zhao et al., 2024). Only one seed was used for the 1B experiment due to compute constraints.

|  | 60M | 130M | 350M | 1B |
|---|---|---|---|---|
| Full | $33.32 \pm 0.22$ (0.36G) | $24.51 \pm 0.03$ (0.76G) | $18.87 \pm 0.18$ (2.06G) | 15.56* (7.80G) |
| LoQT (Ours) | $33.98 \pm 0.15$ (0.23G) | $24.57 \pm 0.01$ (0.49G) | $19.12 \pm 0.01$ (0.98G) | 15.55 (3.16G) |
| LoQT-nq (No quant.) | $33.55 \pm 0.03$ (0.28G) | $24.37 \pm 0.02$ (0.63G) | $18.85 \pm 0.01$ (1.47G) | 15.20 (5.11G) |
| GaLore | $34.15 \pm 0.24$ (0.24G) | $24.81 \pm 0.04$ (0.52G) | $19.47 \pm 0.01$ (1.22G) | 15.64* (4.38G) |
| LoRA | 34.99* (0.36G) | 33.92* (0.80G) | 25.58* (1.76G) | 19.21* (6.17G) |
| ReLoRA | 37.04* (0.36G) | 29.37* (0.80G) | 29.08* (1.76G) | 18.33* (6.17G) |
| $r/d_{model}$ | 128 / 256 | 256 / 768 | 256 / 1024 | 512 / 2048 |
| Training Tokens | 1.1B | 2.2B | 6.4B | 13.1B |

understanding (Wang et al., 2019). Runs were conducted on up to 4x 40GB NVIDIA A100s 2x 80GB NVIDIA H100s, or a single 24GB NVIDIA RTX 3090. The longest run was the training of the 1B models, taking approximately four days on the four A100s. The 3090 was used for throughput and to empirically verify memory claims.

Hyperparameters are consistent across model sizes, with experiments in BF16 format for memory efficiency. All models use a maximum sequence length of 256, a total token batch size of 131K tokens, and a learning rate warmup for the first 10% of the training steps, followed by cosine annealing to 10% of the initial learning rate. Full experimental details, including the specific hyperparameters for each task, are provided in Appendix C.

**Baselines** For pre-training, we compare LoQT against LoRA (Hu et al., 2021), ReLoRA (Lialin et al., 2023), GaLore (Zhao et al., 2024), and non-quantized version of LoQT, LoQT-nq. In our experiments, LoQT, LoRA, and ReLoRA modify attention and fully connected layers while maintaining full-rank embeddings and normalization layers. This contrasts with GaLore, which keeps weights full-rank but projects gradients to low-rank, and standard full-rank training. For fine-tuning, we benchmark LoQT against Ga-Lore, LoftQ (Li et al., 2023), LoRA and LoQT-nq. All models use identical update frequencies for GaLore, ReLoRA, LoQT-nq, and LoQT, starting with an update frequency of $T = 100$ and then with exponentially increasing update frequencies. This means more frequent updates early and fewer as the model stabilizes (see Section 4b for details). All models are trained using the Adam optimizer, except GaLore which uses GaLoreAdam for gradient projection.

### 4.2. Pre-training of Generative Language Models

Results and details for pretraining of language models of sizes 60M, 130M, 350M and 1B parameters are shown in Table 1. Model sizes are calculated based on the full models without any low-rank methods. We see that LoQT and LoQT-nq both perform very close to full rank pretraining and GaLore, while using significantly less memory by keeping most of the model weights in a quantized state. For the 60M model, full training is only slightly better than LoQT, while we see results improve or be within the standard error for the other sizes. We also notice a slight drop in performance from quantizing the original weight matrix, comparing LoQT and LoQT-nq. The key difference between the approaches are the theoretical memory estimates, e.g. where LoQT requires 59% less memory for the 1B model in full precision and 28% less memory than GaLore.

### 4.3. Memory-efficient finetuning

We fine-tune the pre-trained DeBERTa-V3-base[1] (He et al., 2023) model on GLUE tasks using LoQT and compare its performance with a full fine-tuning baseline, LoRA, LoftQ, and GaLore. See Appendix 5 for details on hyperparameters. Results are given in Table 2.

We find that both LoQT-nq and LoQT perform well. Somewhat surprisingly, it sometimes surpasses GaLore, LoftQ, and LoRA. This may indicate that initializing the LoRA factors with information about the gradient of $W$ could be a beneficial starting point compared to standard initialization methods. As the goal of this work is to limit memory consumption, we leave out further comparisons that could verify these findings to future work.

---

[1]From https://huggingface.co/microsoft/deberta-v3-base.

Table 2: Results with LoQT, LoQT-nq, and GaLore of DeBERTaV3-base models on the GLUE development set. We report mean and standard error over three seeds. The best results on each dataset are shown in **bold**.

| Rank | Method | MNLI | QNLI | RTE | SST | MRPC | CoLA | QQP | STSB | Average |
|------|--------|------|------|-----|-----|------|------|-----|------|---------|
| | | Acc | Acc | Acc | Acc | f1 | Matt | f1 | PCorr | |
| 32 | LoQT-nq | 90.0±0.10 | 94.2±0.06 | **84.8±0.75** | **95.9±0.06** | 94.1±0.25 | **72.5±0.41** | **90.0±0.06** | 91.5±0.07 | **89.1** |
| 32 | LoQT | 90.0±0.09 | **94.3±0.04** | 84.1±0.91 | 95.5±0.10 | **94.4±0.20** | 70.5±0.35 | 89.2±0.02 | **91.5±0.13** | 88.7 |
| 32 | LoRA | 89.9±0.03 | 94.0±0.09 | 83.6±0.12 | 95.7±0.15 | 93.5±0.26 | 69.3±0.47 | 89.8±0.11 | 90.7±0.22 | 88.3 |
| 32 | LoftQ | 90.4±0.09 | 93.2±0.02 | 83.8±0.63 | 95.6±0.07 | 93.2±0.14 | 71.1±0.28 | 89.6±0.12 | 91.0±0.09 | 88.5 |
| 32 | GaLore | **90.3±0.07** | 94.0±0.04 | 83.7±0.79 | 95.6±0.07 | 93.4±0.38 | 70.7±0.24 | 89.8±0.05 | 90.6±0.01 | 88.5 |

Table 3: Comparison of memory usage between GaLore, LoRA, and LoQT. $W \in \mathbb{R}^{m \times n}$ ($m \leq n$), rank $r$.

| | GaLore | LoRA | LoQT (Ours) |
|--|--------|------|-------------|
| Weights | $mn$ | $mn + mr + nr$ | $mn + mr + nr$ |
| Optimizer States | $mr + 2nr$ | $2mr + 2nr$ | $2nr$ |
| Gradients | $mn$ | $mr + nr$ | $nr$ |
| Pretraining | Yes | No | Yes |
| Fine-Tuning | Yes | Yes | Yes |
| Quantizeable | No | Yes | Yes |

### 4.4. Memory and Throughput

**Memory usage** An overview of memory usage for Ga-Lore, LoRA and LoQT is given in Table 3. We see that LoQT makes use of the same number of trainable parameters as LoRA for a given rank while using less memory for the optimizer states and gradients than in both LoRA and GaLore.

We compare the LoQT to the closest in memory performance approach, GaLore, for 13B in Figure 1, and for other model-sizes in Figure 6. We compare three different use cases, using the approaches directly, combining them with an 8-bit Adam optimizer (Dettmers et al., 2022b), and using per-layer weight updates with offloading (while still using 8-bit Adam). We see from the figures that LoQT significantly shrinks both the number of trainable parameters and optimizer states compared to GaLore.

Per-layer weight update is essential for GaLore; without it, an additional ∼12 GB of VRAM is needed for gradients in a 7B model, making full-parameter fine-tuning impossible on a 24GB GPU. Additionally, the per-layer gradient updates may not work well with DDP (Distributed Data Parallel) and gradient accumulation. With our method, we can get a lower memory than GaLore even when they use per-layer gradient updates. When not using per-layer gradient updates, this difference becomes even bigger as seen for the 7B model in Figure 6. This allows us to increase the batch size during training on multi-GPU setups, leading to speed improvements when training larger models. We note that the memory required for storing gradients can be reduced in GaLore by doing gradient accumulation in the low-dimensional space and only projecting back once right

before adding the gradient update to the weight matrix. The memory consumption of this approach would fall somewhere between Galore with and without per-layer updates but does still not allow for quantized training.

Moreover, our method supports training 7B models without per-layer computations on 24GB GPU. This makes it possible to use multi-GPU training with gradient accumulation, a capability not possible with the current GaLore approach. Our memory advantage allows for a batch size of 1280 tokens compared to GaLore's 256 for the 7B model on the 24GB RTX3090. With per-layer gradient updates, LoQT can train a 13B model on a single GPU, pushing the limits of hardware efficiency.

**Throughput** We evaluate the throughput with a sample batch size of 16 with a total batch size of 512 using gradient accumulation, which is the largest power of 2 that fits on the GPU. We update the projection matrix $P$ every 200 iterations. The per-layer gradient update algorithms apply a weight update for every mini-batch as they do not support gradient accumulation. For evaluation, we use a 1B parameter model with rank 512. We find that LoQT can process 16% fewer tokens per second than only using AdamW, at 3996 tokens/s compared to 4782 tokens/s on the RTX3090.

## 5. Ablations

**Quantization Error Compensation and Initialization** To assess the impact of quantization error compensation, we analyze the validation loss curves for a 130 million parameter model. Figure 4a shows that quantizing $W$ or both $W$ and $P$ without error compensation, or exponential frequency updates, causes the loss to stagnate early. We also note that quantizing $P$ has a much smaller effect on the loss compared to quantizing $W$. Error compensation significantly improves the model's performance, resulting in approximately 3.5 points better perplexity. Adding exponentially increasing update frequency improves perplexity by an additional 1.5 points, achieving performance close to that of models without quantization.

Without the quantization error compensation detailed in

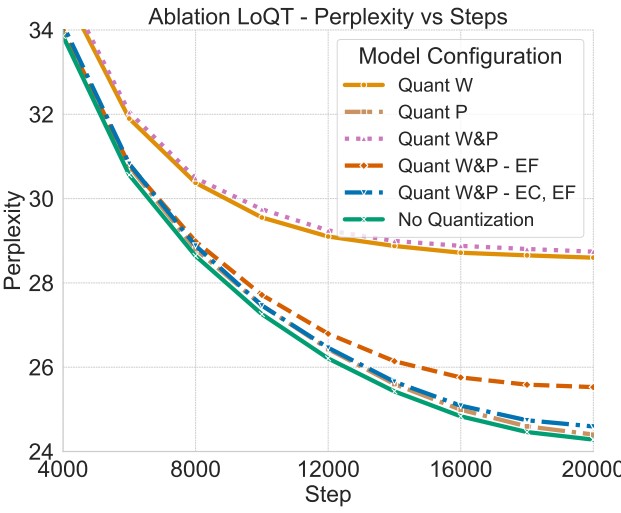

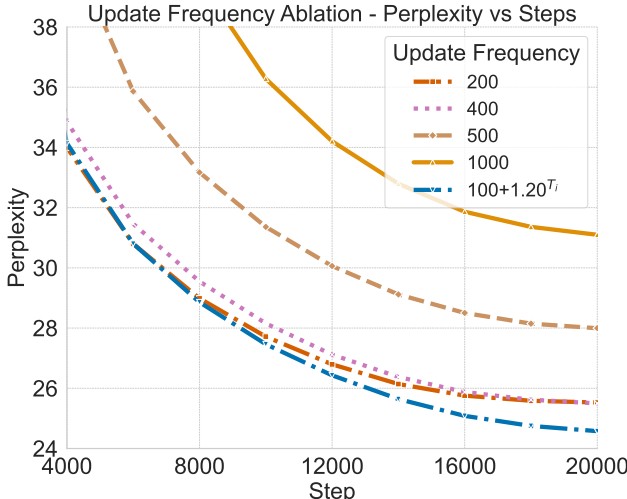

(a) Ablation of quantization effects, with/without EC (Error Compensation) and EF (Exp. Decreasing Update Frequency).

(b) Ablation of fixed update frequencies (200, 400, 500, 1000) and exponentially increasing frequency starting at 100 (factor 1.2).

Figure 4: Ablation results for update frequency, error-compensation, quantization, model size 130m, and rank 256.

§3.5, LoQT's performance stagnates earlier and diverges more from the other models. This demonstrates the effectiveness of our compensation approach in mitigating the quantization errors introduced during the $W$ update with $AB$ and subsequent quantization steps.

**Projection update frequency**  Our scheduling approach ensures frequent updates early in training for substantial weight adjustments, with decreasing frequency as training progresses. This allows for larger updates to compensate for smaller ones canceled out by quantization errors. Figure 4b presents an ablation study on our method of progressively increasing update frequency starting at 100 and increasing by a factor of $1.2^T$ up to 2500. We show validation loss curves for fixed update frequencies 200, 400, 500, and 1000.

The results show that exponentially increasing the update gap is particularly beneficial for models employing quantization, enabling them to achieve the same perplexity as those without quantization while making use of GaLore. Conversely, the performance gains are more subtle for models that do not use quantization and rely solely on GaLore. This could be due to the reduction in the accumulation of errors from frequent updates of the projection factor $P$, as the influence of outdated optimizer statistics becomes less prevalent. Finally, an ablation on the ranks used for $P$ and $B$ is given in Figure 5 in the Appendix.

## 6. Discussion and Conclusion

We present LoQT, a method for memory-efficient pretraining and adaptation of quantized models. The key insights behind the approach are the benefits of initializing two low-rank factors using the gradient of the weight matrix and us-

ing exponentially increasing update gaps. By training only one low-rank factor in higher precision, we keep the other factor and the main weight matrix frozen and quantized. We then periodically merge in updates from the trainable factor into the quantized original matrix, allowing the frozen matrix to store updates while the trainable factor guides optimization based on gradient information.

While our initial goal was to lower memory usage to facilitate the training of models such as LLMs on consumer-grade hardware, we are cautiously excited about the results sometimes being better than the baselines. These evaluations will be explored in more detail in future work.

Our method is general and opens up new ways of decreasing memory use as well as improving the training throughput. This could be done by implementing kernel fusion and using other quantization methods such as NF2 (Dettmers et al., 2023a) or quantization of activations, making it possible to do the matrix multiplications using modern tensor core formats such as FP8 or INT4.

## 7. Impact and Limitations

Our work has the potential to have a significant impact on those working in hardware-constrained settings by enabling more efficient training on consumer hardware. We are particularly excited to see the method being applied in single GPU settings. We validate LoQT on several model sizes, by training over many steps and by fine-tuning on a standard benchmark for natural language understanding. While we are confident in our results, further exploration of training duration, data diversity, and hyper-parameter tuning might lead to different results in those settings.

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

# A. Quantization Methods

Quantization methods can be broadly categorized into Quantization-Aware Training (QAT), Post-Training Quantization (PTQ), and Fully Quantized Training (FQT).

**Quantization-Aware Training (QAT)** QAT (Liu et al., 2023; Jung et al., 2018; Egiazarian et al., 2024; Wang et al., 2023; Ma et al., 2024) integrates quantization in the training process by emulating inference time quantization where the model weights are quantized. By maintaining high precision gradients and optimizer states, QAT allows the model to adapt to quantized weights while preserving accuracy. These methods predominantly focus on weight-only quantization approaches, which involve converting weight matrices into low-precision formats and then upcasting them just before computation (Wang et al., 2023; Ma et al., 2024). This allows the main computation to occur at high precision, effectively preserving model accuracy while significantly compressing the model (Frantar et al., 2023). However, QAT can require significant computational resources due to the need for full precision gradient calculations and large optimization states (Dettmers et al., 2023a).

**Post-Training Quantization (PTQ)** PQT (Frantar et al., 2023; Dettmers et al., 2023b; Tseng et al., 2024; Xiao et al., 2024; Park et al., 2024; Dettmers et al., 2022a; Lee et al., 2023; Heo et al., 2024; Shao et al., 2024) involves converting a pre-trained high-precision model into a lower precision format. This can be done directly or by using a subset of the training data to calibrate the quantization process or fine-tune the quantized weights to adapt the model to the quantization. However, PTQ often results in reduced accuracy compared to QAT because the model does not learn to adjust to quantized weights during training (Frantar et al., 2023; Xiao et al., 2024).

**Fully Quantized Training (FQT)** FQT aims to minimize memory and accelerate the training process by quantizing both forward and backward passes (Wang et al., 2018b; Chmiel et al., 2022; Banner et al., 2018; Perez et al., 2023; Wortsman et al., 2023). These methods often require specialized hardware (Peng et al., 2023; Xi et al., 2024) but are promising for efficient training, and current approaches cannot maintain accuracy (Xi et al., 2024).

LoQT is a form of QAT that gets close to FQT. As we perform a variant of LoRA (see §2.2), we factor the layers $W$ into two matrices $P$ and $B$. We quantize the $W$ and $P$ with NF4, but keep $B$ in 16-bit precision. We periodically update the $W$ matrices using the product of the fixed $P$ and the updated $B$s without ever dequantizing it all at once, only layerwise when merging in $PB$. This approach retains the benefits of reduced memory usage while minimizing accuracy loss, focusing high-precision updates on a low-rank representation, and allowing efficient model updates without the overhead of full matrix re-quantization.

# B. Intuition for Only Training the $B$ Adapter and Keeping $P$ Fixed

The intuition behind training only the $B$ matrix while keeping $P$ fixed is that $P$ defines a low-rank subspace for optimization, containing information about the direction of optimization for the full model. By updating only $B$, we ensure all updates remain within this predefined subspace. This approach reduces the memory required due to only storing gradients and optimizer states for $B$. Additionally, by keeping $P$ fixed it can be quantized for further memory savings. Experiments also showed that optimizing both $B$ and $P$ led to less consistent convergence than only optimizing $B$.

# C. Hyperparamters

### C.1. Pre-training

For the pre-training results shown in Table 1, we adopted configurations from GaLore (Zhao et al., 2024) and tested pre-training methods on different LLaMA 2 model sizes using the C4 dataset. Training was conducted with optimizer states in BF16 precision, and NF4 precision quantization was used for LoQT. The model rank was adapted based on the largest layer with specific parameters.

Table 1 shows the ratio $r/d_{model}$, which denotes the rank relative to the largest weight matrix dimension. All experiments used a maximum sequence length of 256, learning rate warmup for the first 10% of training steps, and cosine annealing for the learning rate schedule, decaying to 10% of the initial rate.

Galore, LoQT-nq, and LoQT used exponentially increasing update frequencies starting at 100 and increasing by $100 + \psi^i$,

where $\psi$ is 1.2 and $i$ is the update counter (see Section D.1 for more details).

We tested learning rates of 0.01, 0.005, 0.001, and 0.0005 across different model sizes. For models ranging from 60M to 350M parameters, a learning rate of 0.01 yielded the best performance. In contrast, full-rank models required smaller learning rates: 0.001 for 60M to 350M models and 0.0005 for the 1B model. To scale the learning rates for LoQT, LoQT-nq, and Galore, we employed a scale parameter $s$ set to 0.5 and 0.25 for the 1B model. This parameter functions similarly to the LoRA alpha parameter, determining the weight on the learned factors for LoQT and LoQT-nq. For Galore, our experiments indicated that $s = 0.5$ was more effective than the 0.25 reported in (Zhao et al., 2024). This scaling approach effectively adjusts the learning rate, resulting in an actual rate of 0.005 for the multi-head attention and feed-forward layers in LLaMA models, which is relatively large compared to the 0.001 used for full-rank models. Higher learning rates led to spikes in the training loss for both full-rank and LoQT models.

Table 4: Pre-training hyperparameters of LLaMA models for evaluation. (-) Indicates we did not train such a model.

| Model Size | Hidden/Intermediate | Attention Heads | Layers | Steps | Data Amount | Rank |
|:---:|:---:|:---:|:---:|:---:|:---:|:---:|
| 60M | 512 / 1376 | 8 | 8 | 10K | 1.3B | 128 |
| 130M | 768 / 2048 | 12 | 12 | 20K | 2.6B | 256 |
| 350M | 1024 / 2736 | 16 | 24 | 60K | 7.8B | 256 |
| 1B | 2048 / 5461 | 24 | 32 | 100K | 13.1B | 512 |
| 7B | 4096/11008 | 32 | 32 | - | - | 1024 |
| 13B | 5120/13824 | 40 | 40 | - | - | 1024 |

### C.2. Fine-tuning

We test learning rates in the range of $1 \times 10^{-5}$ to $5 \times 10^{-4}$. For LoQT LoftQ, we employed normal float NF4 quantization and performed 5 iterations of optimizing the error of quantization. We used a batch size of 32 and a maximum sequence length of 256. Table 5 summarizes the detailed hyperparameters for tasks in GLUE using the DeBERTaV3-base model. We use a fixed projection gap of 2400 for all runs.

Table 5: Hyperparameter setup for LoQT-nq, LoQT, LoftQ (Li et al., 2023), LoRA (Li et al., 2023), and Galore across various tasks on the GLUE benchmark.

| Method | Hyper-parameter | MNLI | RTE | QNLI | MRPC | QQP | SST-2 | CoLA | STS-B |
|:---:|:---:|:---:|:---:|:---:|:---:|:---:|:---:|:---:|:---:|
| LoQT, LoftQ | # Epochs | 5 | 20 | 10 | 60 | 10 | 10 | 20 | 60 |
| | Learning Rate | $1 \times 10^{-4}$ | $5 \times 10^{-5}$ | $5 \times 10^{-5}$ | $1 \times 10^{-4}$ | $5 \times 10^{-5}$ | $5 \times 10^{-5}$ | $1 \times 10^{-4}$ | $5 \times 10^{-5}$ |
| LoRA, Galore | # Epochs | 10 | 30 | 30 | 30 | 30 | 30 | 30 | 30 |
| | Learning Rate | $1 \times 10^{-5}$ | $2 \times 10^{-5}$ | $1 \times 10^{-5}$ | $2 \times 10^{-5}$ | $1 \times 10^{-5}$ | $2 \times 10^{-5}$ | $2 \times 10^{-5}$ | $3 \times 10^{-5}$ |

## D. Rank Ablation

Figure 5 shows the validation perplexity versus training steps for various ranks using LoQT-nq and LoQT on a 130 million parameter model over 20,000 iterations. All models employ an exponentially increasing update frequency starting at 100, with a factor of $1.2^{T_i}$. The results demonstrate that both the quantized (LoQT) and non-quantized (LoQT-nq) models follow a similar trajectory for ranks ranging from 64 to 512. However, for the smaller rank of 64, there is a slight divergence between LoQT-nq and LoQT, indicating a limit to how low the rank can be while maintaining accuracy with quantization. This plot highlights the tradeoff between rank and perplexity, suggesting that while our method supports low-rank training, there is a minimum rank threshold needed to achieve results comparable to regular pre-training.

### D.1. Memory Measurements

Figure 6 demonstrates that LoQT requires less memory than GaLore and Adam, even without using per-layer gradients (Lv et al., 2023) or Adam 8-bit (Dettmers et al., 2022b). The gap between LoQT and the baselines increases with larger model sizes. The configurations and ranks for each model are shown in Table 4. With LoQT and Adam 8-bit, it is possible to

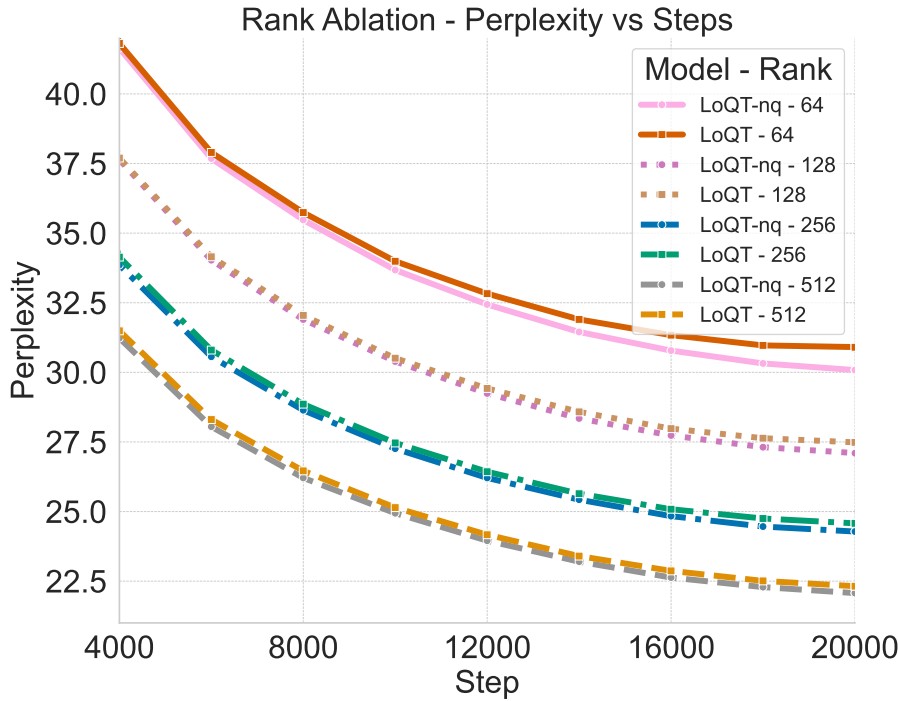

Figure 5: Rank ablation for LoQT and LoQT-nq showing perplexity as a function of steps.

pre-train a 13B model with rank 1024 on a GPU with 24GB of VRAM. This enables training with LoQT on consumer GPUs, such as the NVIDIA 4090, using a small per-GPU token batch size 256. Figure 1 in the main text provides a detailed breakdown of each memory component for the 13B model. Maximum memory allocated is measured using `nvitop` (`https://github.com/XuehaiPan/nvitop`).

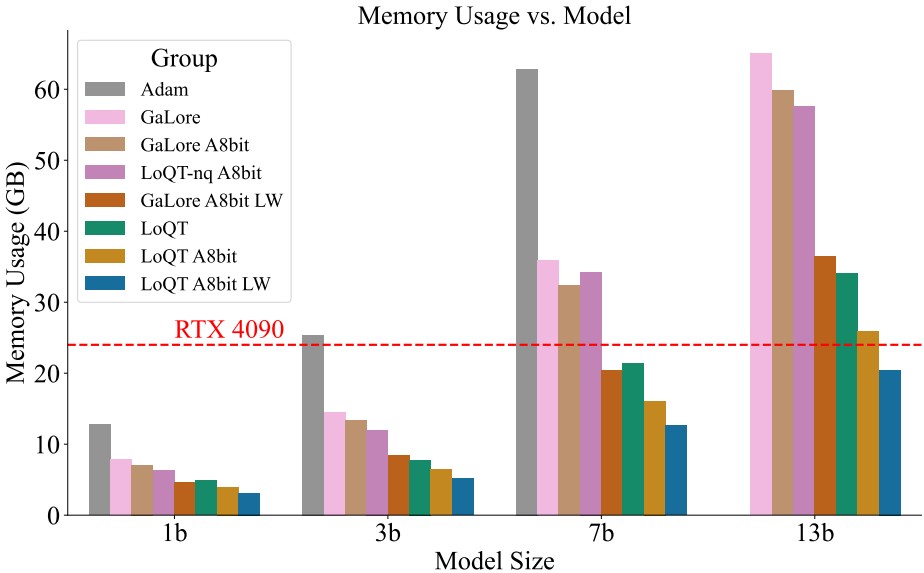

Figure 6: Memory usage for LoQT vs baselines for different model sizes. LW means per-layer gradient updates as per (Lv et al., 2023), and A8bit mean with Adam 8-bit. We evaluate using a token batch size of 256.

