# OpenReview forum: "LoQT: Low Rank Adapters for Quantized Training"
_ICML.cc/2024/Workshop/WANT — WANT@ICML 2024 Oral_

### Official Review · Reviewer_GedX · 2024-06-11
**LoQT: a combination of QLoRA and GaLore methods for low-memory pretraining and fine-tuning**

**Confidence:** 4

**Summary:**

The LoQT paper is building on top of recent quantization and low rank methods to present a novel method for pretraining and fine-tuning algorithm, enabling LLM such as Llama 13B to fit entirely on a single consumer GPU while matching accuracy of higher precision and full rank training.

The main idea of the paper is to re-use GaLore [1] low-rank gradient projection and combine it with the now common LoRA weight decomposition. As noticed by the authors, on every time interval GaLore projection matrix is constant, one does not need to update the full weight tensors of a model, but can just update the low-rank adapter. While this aspect in itself does not reduce memory footprint during training (as full weights still need to be kept), it can be combined with QLoRA: the main weights can be efficiently quantized to 4bits using the NF4 data format, reducing the model footprint by a factor 2. Finally, combined with 8-bit Adam optimizer techniques, the authors show that Llama 13B could be trained on a single GPU with 24GB of memory.

[1] https://arxiv.org/pdf/2403.03507
[2] https://arxiv.org/abs/2305.14314

**Strengths:**

The main strength of the paper is to present a novel training scheme optimizing memory footprint on all fronts (i.e. model state, optimizer state and gradients), while maintaining the same accuracy as full rank + full precision training. It builds elegantly on top of GaLore and QLora methods. The authors have extensive experiments (as well as ablation studies) on different model size to show the robustness of their method.

LoQT has the potential of being widely adopted by the machine learning communities as it helps lowering the hardware compute budget necessary for pretraining and fine-tuning LLMs.

**Weaknesses:**

The main (small) weakness of the paper is the potential brittleness of the projection update frequency, i.e. $100 + 1.2^T$. The ablation study shows that a scheme more complex than a constant update frequency is necessary, but the downside is then the introduction of additional scheduling mechanism (and hyper-parameters associated) in the training scheme, on top of the classic learning rate schedule.
It could be potentially interesting to investigate if it can be replaced that by a more "dynamic" rule, checking that the update $B_{T-1}$ is above a certain level of NF4 quantization noise to trigger the main weight $W_T$ update. From the perspective and experience of low precision training literature, it would feel like a more robust approach than a pre-determined scheduling rule.


On the presentation aspect, I believe GaLore weaknesses are slightly mis-represented and overstated in Section 4.4. It is fairly easy to apply GaLore gradient projection $P^T G_t$ directly inside the backward pass of a model, meaning using GaLore leads to gradient memory savings even when per-layer update is not applied. And additionally, it also means that it can be combined efficiently with gradient accumulation or/and DDP, as these methods can be directly done in the low-rank gradient space (the projection being a linear operator).

---

### Official Review · Reviewer_thmZ · 2024-06-13
**Nice work on the parameter efficient training for quantized models.**

**Confidence:** 3

**Summary:**

This paper proposes LoQT which is suitable for quantized models for both pretraining and finetuning. The method iteratively updates the weight matrix. As a result, LoQT achieves better compression and in the meanwhile the best performance in eval.

**Strengths:**

1. The paper is clearly written and the method is explained in details.
2. Quantitative analysis and comparison with other LoRA variants are very comprehensive.

**Weaknesses:**

1. The method is a bit more complicated than vanilla lora since it composes iterative merge/refactoring steps.
2. What's the intuition of not updating P during the training cycle?

---

### Official Review · Reviewer_HumH · 2024-06-14
**LoQT: Low Rank Adapters for Quantized Training**

**Confidence:** 3

**Summary:**

The study introduces a novel language model training method Low Rank Adapters for Quantized Training (LoQT). This method combines ideas from GaLore and quantization, enabling the training of models with 13 billion parameters on consumer-grade GPUs. In LoQT, the weight update is decomposed into low-rank matrices P and B. P is initialized using the SVD decomposition of the gradient of the weight, and B is initialized to reduce the quantization error. Only B is trained in LoQT. PB is merged back to the full rank matrix W after certain update steps. This process continues until the training stops. LoQT performs better than GaLore in pre-training language models while saving memory. This observation is evident from 60M to 1B parameter models. LoQT performs better than GaLore and LoRA in finetuning when trained and evaluated on GLUE.

**Strengths:**

- The paper is well-written, with clear explanations of the motivation, method, and results, making it easy to follow.
- LoQT effectiveness in pre-training and fine-tuning is supported by extensive experiments.
- Theoretical justification is provided for the claims, including the derivation on how P.T G can be replaced by B.
- The study includes numerous ablation studies, providing a comprehensive understanding of the method.
- The method allows fitting a 13B parameter model on a single GPU, an important step toward memory-efficient pre-training of large language models.
- Overall, this study is solid and impactful.

**Weaknesses:**

- The motivation and derivation to initialize the B matrix with P^(-1)(W_q - W) can be explained more clearly and in detail.
- The paper would benefit from insights on why LoQT-nq uses more memory than GaLore in pre-training models.

---

### Meta-Review · Area_Chair_1jYL · 2024-06-16

**Recommendation:** Accept (Oral)
**Confidence:** 4

**Metareview:**

## Strengths
* This paper is well written and the methods are clearly explained and motivated
* The paper contains both extensive experiments and a theoretical justification
* This approach enables training a 13B parameter on a single 24GB GPU, which has a high potential
  for a strong impact in the community

## Weaknesses
* Some specific points could be explained more clearly
* This approach adds a new hyper-parameter for projection update frequency, and it is unclear what the best
  value is. Maybe a dynamic approach would be more generalizable to other contexts/neural networks

The general sentiment is very largerly positive, I recommend acceptance as an oral presentation.

---

### Decision · Program_Chairs · 2024-06-17

**Decision:**

Accept (Oral)

**Comment:**

We thank the authors for their time and contribution to WANT and we are pleased to share that after the reviewing process the paper has been accepted. Congratulations! We encourage the authors to consider reviewers' feedback for the improvement of the camera-ready version. We hope to see you in person at the workshop and brainstorm on efficient training research together!